# Impact of circ-0000221 in the Pathogenesis of Hepatocellular via Modulation of miR-661–PTPN11 mRNA Axis

**DOI:** 10.3390/pharmaceutics14010138

**Published:** 2022-01-06

**Authors:** Marwa Matboli, Mohmed Kamal Hassan, Mahmoud A. Ali, Mohamed Tarek Mansour, Waheba Elsayed, Reham Atteya, Hebatallah Said Aly, Mahmoud El Meteini, Hesham Elghazaly, Sherif El-Khamisy, Sara H. A. Agwa

**Affiliations:** 1Medical Biochemistry and Molecular Biology Department, Faculty of Medicine, Ain Shams University, Abbassia, Cairo 11381, Egypt; hebatallahsaid@med.asu.edu.eg; 2Center for Genomics, Helmy Institute for Medical Sciences, Zewail City for Science and Technology, Giza 12578, Egypt; mfarah@zewailcity.edu.eg (M.K.H.); p-walsayed@zewailcity.edu.eg (W.E.); p-rattaeya@zewailcity.edu.eg (R.A.); 3Biotechnology Program, Biology Division, Zoology Department, Faculty of Science, Port Said University, Port Said 42526, Egypt; 4Department of Biomedical Research, Armed Forces College of Medicine (AFCM), Cairo 11774, Egypt; mahmoudali@gmail.com (M.A.A.); or mohammadtarek459@gmail.com (M.T.M.); 5Department of General Surgery, The School of Medicine, University of Ain Shams, Abbassia, Cairo 11382, Egypt; mahmoud_elmeteini@med.asu.edu.eg; 6Oncology Department, Faculty of Medicine, Medical Ain Shams Research Institute (MASRI), Ain Shams University, Cairo 11382, Egypt; heshamelghazaly@med.asu.edu.eg; 7The Healthy Lifespan Institute, The Institute of Neuroscience, Department of Molecular Biology and Biotechnology, University of Sheffield, Sheffield S10 2TN, UK; s.el-khamisy@sheffield.ac.uk; 8The Institute of Cancer Therapeutics, West Yorkshire BD7 1DP, UK; 9Clinical pathology and Molecular Genomics Unit, Faculty of Medicine, Medical Ain Shams Research Institute (MASRI), Ain Shams University, Cairo 11382, Egypt

**Keywords:** hepatocellular carcinoma, hsa-circ-0000221, miR-661, PTPN11

## Abstract

Hepatocellular carcinoma (HCC) is a leading cause of cancer-related death in Egypt. A deep understanding of the molecular events occurring in HCC can facilitate the development of novel diagnostic and/or therapeutic approaches. In the present study, we describe a novel axis of *hsa-circ-0000221*–miR-661–PTPN11 mRNA proposed by in silico and in vitro analysis and its role in HCC pathogenesis. We observe a reduction in the expression levels of *hsa-circ-0000221* and PTPN11 mRNA in HCC patients’ sera tested compared with control subjects. The reduction occurs with a concomitant increase in the expression of miR-661. Furthermore, the introduction of exogenous *hsa-circ-0000221* into Hep-G2 or SNU449 cell lines results in detectable decrease in cellular viability and an increase in apoptotic manifestations that is associated with G1 accumulation and CCDN1 overexpression. Altogether, these findings indicate the tumor-suppressive role of *hsa-circ-0000221* in HCC, which acts through miR-661 inhibition, along with a subsequent PTPN11 mRNA increase, where PTPN11 is known to inhibit cell proliferation in many forms of cancer. Our study encourages further investigation of the role of circRNAs in cancer and their potential use as molecular biomarkers.

## 1. Introduction

Hepatocellular carcinoma (HCC) is the most common type of liver cancer and one of the major causes of cancer-related death worldwide [1]). Early diagnosis of HCC is possible, and radical surgeries are a common treatment modality with mostly favorable prognostic outcomes [2]. However, many patients with HCC are missed with the current diagnostic standards and, therefore, lose the operation opportunity. To improve the therapeutic outcome and prognosis of HCC patients, it is imperative to search for more efficient biomarkers to increase the early diagnosis rate of HCC. Many HCC patients respond poorly to sorafenib (the first-line chemotherapeutic therapy for advanced HCC) and develop drug resistance after several months of treatment. Chemotherapeutic resistance in HCC involves different mechanisms, e.g., epithelial–mesenchymal transition, cancer stem cells, autophagy, and epigenetic regulation [3]. Several signaling pathways are enrolled in chemotherapeutic resistance in HCC such as TNFα/NF-κB, Wnt/β-catenin, TGFβ, Ras/MEK/ERK, and JAK/STAT pathways [4].

Circular RNAs (circRNAs) are a class of noncoding RNAs (ncRNAs) characterized by their covalently closed structure formed by the end-to-end joining of RNA transcripts known as “back splicing” [5] CircRNAs display high levels of stability compared to their ncRNA counterparts, are highly represented in the eukaryotic transcriptome, and often show tissue-/developmental stage-specific expression [6]. They are even present at higher concentrations than messenger RNAs (mRNAs) in the cytoplasm of certain cells [7]. CircRNAs can inhibit the binding of miRNAs to mRNAs through “sponging”, thus regulating the expression of their downstream target genes. circRNAs may act as competing endogenous RNA (ceRNA) via modulating miRNA expression, leading to NF-κB activation that may result in cancer progression, as well as mediate chemo resistance [8].

CircRNAs are strongly associated with tumors [9] and have previously been shown to play a role in the development and progression of HCC [10]. For example, circRNA-7 has an inhibitory effect on miR-7 via sponging. miR-7 is a tumor-suppressor miRNA demonstrated through several mechanistic studies to be often expressed in relatively low levels in various tumor tissues and is negatively correlated with the growth, invasion, and colony formation of these tumors through regulating different target molecules [11]. miR-7 has also been found to arrest cell cycles at the G0/G1 phase and inhibit the spread and metastasis of HCC through downregulating three target molecules (PIK3CD, mTOR, p70S6K) belonging to the PIK3CD/Akt signaling pathway [12]. circRNA-SORE (a circular RNA upregulated in sorafenib-resistant HCC cells) plays a crucial role in the maintenance and spread of sorafenib resistance [4].

In the current study, we used in silico data to uncover the *hsa-circ-0000221*–miR-661–PTPN11 mRNA axis, relevant to HCC with significant differential expression, followed by validation in HCC cells and in clinical samples representing a small pilot study to examine their expression levels in serum samples.

## 2. Materials & Methods

### 2.1. In Silico Analysis and Prediction

Circ2Traits (Available at http://gyanxet-beta.com/circdb/, Accessed on 23 November 2019) [13] and CircNet (Available at http://circnet.mbc.nctu.edu.tw/, Accessed on 23 November 2019) [14] bioinformatics databases were used for retrieval of HCC-specific circRNA. Afterward, HCC-relevant miRNA targets of circular RNA *hsa_circ_0000221*, as well as their binding sites on the circular RNA, were predicted using the CircInteractome tool (Available at http://circinteractome.nia.nih.gov/, Accessed on 23 November 2019) [15], The mRNA targets of hsa-mir-661 were predicted using TargetScan (Available at http://www.targetscan.org/) [16] and miRWalk (Available at zmf.umm.uni-heidelberg.de/apps/zmf/mirwalk2/, Accessed on 23 November 2019) [17] databases. The network of circRNA–miRNA–mRNA was constructed using Cytoscape software (Version 3.5.1) (Available at http://www.cytoscape.org/, Accessed on 23 November 2019) [18]. Lastly, the protein–protein interactions of PTPN11, as well as their pathway enrichment, were studied through the STRING database (Available at http://string-db.org/, Accessed on 23 November 2019) [19].

To gain insights into the role of *hsa-circ-0000221* in HCC, the predicted miRNA targets of the circular RNA were subjected to pathway enrichment analysis using the miRPath tool (Available at http://snf-515788.vm.okeanos.grnet.gr/, Accessed on 23 November 2019) [20]. Additionally, the predicted target mRNAs of hsa-mir-661 were subjected to gene ontology prediction through GOSlim on the Web Gestalt database (Available at http://www.webgestalt.org, Accessed on 23 November 2019) [21]. Subsequently, the Enrich tool (Available at http://amp.pharm.mssm.edu/Enrichr/, Accessed on 23 November 2019) was used to predict the significant pathways where hsa-mir-661 targets could be enriched. To predict the functional implications of PTPN11 regulation in HCC and its targeting by miRNAs, the miRWalk database [17] was used. Additionally, the STRING database [22] was used to analyze the protein–protein interactions of PTPN11 and investigate the enriched KEGG pathways.

### 2.2. Antibodies and Reagents

The following antibodies were used for Western blotting: mouse anti-human BAX mAb, cyclin D1 mAb, cyclin B1 mAb, (BD Bioscience; Erembodegem, Belgium), procaspase-3 mAb, active caspase-3 mAb, PARP mAb (Santa Cruze, CA, USA), and rabbit anti-human GAPDH pAb (Abcam, Cambridge, UK). All were used at a dilution of 1:200, except GAPDH which was used at a dilution of 1:1000. As for immunoblotting detection, either anti-mouse or anti-rabbit secondary horseradish peroxidase-conjugated antibodies (Dako, Glostrup, Denmark) were used and diluted at 1:1000.

### 2.3. Cell Lines

The human hepatocellular carcinoma cell lines used in this study, HepG2 and SNU449, were obtained from ATCC. HepG-2 cells were maintained in Dulbecco’s modified Eagle’s medium (DMEM, Gibco-BRL, Gaithersburg, MD, USA) while SNU449 cells were maintained in RPM1640 medium (Sigma, St. Louis, MO, USA). The media of both cells contained 10% heat-inactivated fetal bovine serum (FBS, Gibco-BRL), 1% l-glutamine, and 1% penicillin/streptomycin (Gibco-BRL), and they were cultured at 37 °C in a 5% CO_2_ atmosphere. HepG2 and SNU449 cells were used from passage number 40 to 51 for the experiments in this study. For transfection assays, cells were seeded to be around 60–70% confluency on the day of transfection.

### 2.4. Human Samples

This study was approved by the institutional review committee board of Ethics, Ain Shams University, Faculty of Medicine, Egypt, under the Code of Ethics of the World Medical Association (Helsinki Declaration) with the approval number FMASU MD 32/2016. Informed written consent was obtained from every participant. Participants were recruited from the clinic of Tropical Medicine Department, Ain Shams University Hospital in the period from January to April 2016. All patients who had received radiation, chemotherapy or surgical intervention were excluded from the study. Demographic data of the population of patients in this study are summarized in Table 1. Serum samples from 25 histologically confirmed HCC patients, 15 hepatitis C virus (HCV) patients, and 10 healthy normal participants were used for total RNA isolation using miRNeasy Mini Spin Columns (Qiagen, Hilden, Germany).

### 2.5. QRT-PCR for circRNA–miRNA–mRNA Genetic Network in Human Serum Samples

The RT2 miRNA First-Strand Kit (Qiagen, Valencia, CA, USA) using miScript HiSpec buffer was used to prepare cDNA from 1 µg of RNA. Real-time PCR was performed on the Step One Plus™ System (Applied Biosystems Inc., Foster, CA, USA). All reactions were performed in triplicate.

The *hsa-circ-000520* and *PTPN mRNA* (NM_002834) expression in HCC cell lines was assessed using QuantiTect SYBR Green PCR Kit (Qiagen, Valencia, CA, USA) and gene-specific primers (Circular RNA specific Quantict Primer and Hs_PTPN11_1_SG Quantict Primer), on a Step One Plus™ System (Applied Biosystems Inc., Foster, CA, USA). Beta-actin was used as a housekeeping gene. Divergent primers for *hsa-circ-000520* were designed by the Circinteractome database (Available at http://circinteractome.nia.nih.gov/, Accessed on 23 November 2019) and synthesized by Qiagen (Hilden, Germany) (forward primer: AGCGCACCTTGTCGATGTAG; reverse primer: AGAATAAGATCCTGCTGGCCG).

miR-661 expression in HCC cell lines was assessed by mixing the total cDNAs with a miRNA-specific forward primer (miScript Primer Assay) (accession: MIMAT0003324) and miScript SYBR Green PCR Kit (Qiagen/SABiosciences Corporation, Frederick, MD, USA) according to the manufacturer’s protocol. RNU-6 was used as an internal control.

The PCR program for SYBR green-based qPCR was as follows: denaturation at 95 °C for 15 min, followed by 40 cycles of denaturation for 10 s at 94 °C, then annealing for 30 s at 55 °C, and a final extension for 34 s at 70 °C. Each reaction was done in duplicate. Relative quantification of gene expression was calculated using the 2^−ΔΔCt^ method [23]. The cycle threshold (Ct) value of each sample was calculated using Step One Plus™ software v2.2.2 (Applied Biosystems). Any Ct value above 36 was considered negative. Amplification plots and T_m_ values were checked to ensure the specificities of the amplicons.

### 2.6. Viability Assay

Cell viability was assayed utilizing a cell counting kit (CCK-8; Dojindo, Kumamoto, Japan). A 96-well plate containing precultured cells (3000 cells/well) was subjected to medium replacement by the WST-8 reagent (2-(2-methoxy-4-nitrophenyl)-3-(4-nitrophenyl)-5–2,4-disulphonyl)-2*H*-tetrazolium monosodium salt) (which, when reduced, turns to orange formazan) at the indicated timepoints. Both the developed color and the absorbance were measured at 450 nm using a microplate reader (FLUOstar Omega, BMG labtech, Ortenburg, Germany), with the amount of formazan being directly proportional to the number of living cells.

### 2.7. Clonogenic Assay

SNU449 and HepG2 cells were cultured in the proper medium (at 50–60% confluence). The following day, cells were transfected with either the *hsa-circ-0000221* expressing vector or the mock. One day after transfection, cells were trypsinized and re-cultured at 400 cells/3 cm dishes. Two weeks later, colonies were fixed by 70% methanol and stained with Giemsa, and pictures were taken for at least three pairs of dishes from each different experiment. Colonies were counted and the survival fraction was calculated relative to the vector control.

### 2.8. Plasmid Construction and Transfection

We generated *hsa-circ-0000221* cDNA using serum samples. Then, 2 µL of RNA was reverse-transcribed by QIAGEN OneStep RT-PCR Kit (Qiagen, Valencia, CA, USA). The divergent primers for *hsa-circ-000520* were used for PCR and generated a PCR product with a size consistent with the target fragment and that published on CircBase. The *hsa-circ-0000221* fragment was cloned into T/A cloning vector using T/A cloning kit (Invitrogen, Carlsbad, CA, USA). and then subcloned into pcDNA™ 3.1(+) vector (Invitrogen, Carlsbad, CA, USA) using T4 DNA. The target fragment was sequenced to determine its full length and to confirm the back-splice junction of *hsa-circ-0000221*: 5′–AGGTCCCCCAGGCGCGACTTGCCTTGGCCCTTGAGCTGCTCGAGCTCGGCCAGCAGGATCTTATTCTGCTGCTCCAGGAAGCGCACCTTGTCGATGTAGTTGGCGAAGCGGTCATTCAGCTCCTGCAGCTCCACCTTCTCGTTG–3′.

The construct was introduced into cells using Lipofectamine^TM^ 2000 reagent (Invitrogen, Carlsbad, CA, USA) according to the manufacturer’s instructions. Then, 1 or 2 days after transfection, at least two million HCC cells were used for isolation of total RNA including miRNAs followed by qRT-PCR to detect the expression of the circRNA-associated ceRNA network (as previously described). For co-transfection, the same plasmid was used together with GFP expressing vector at a ratio of 3:1 (circRNA:GFP)

### 2.9. Flow Cytometry Analysis

After transfection with the target plasmid or mock, phosphate-buffered saline (PBS) was used to wash the trypsinized cells twice, followed by cell-cycle phase analysis. This was done by fixing the cells overnight at 4 °C in 70% ethanol. Cells were washed with Ca^2+^/Mg^2+^-free Dulbecco’s PBS and treated with 1 mg/mL RNase (Type I-A; Sigma, St. Louis, MO, USA). Treated cells were then stained using 10 μg/mL propidium iodide (PI; Sigma) for 20 min; then, the cells were filtered and kept on ice until measurement. Cells were obtained using the FACS calibrator using nontreated stained cells. In this study, we quantified and took into consideration cell fractions with a DNA content lower than G0/G1, the sub-G0/G1peak.

### 2.10. Annexin V Staining

Two days after transfection with the target plasmid, cells were harvested and washed with PBS. Then, cells were stained directly with PI at a final concentration of 10 μg/mL and 2% annexin V Flous (BioRad, Hercules, CA, USA) in incubation buffer (10 mM HEPES/NaOH, pH 7.4, 140 mM NaCl, 5 mM CaCl_2_) for 10 min. Cells were acquired using the FACS calibrator using nontreated stained cells as control. We then quantified cells stained with annexin V (early and middle apoptotic) and those stained with PI (late death). FLowJo software was used to analyze Apoptotic cells.

### 2.11. Western Blotting

RIPA buffer (10 mM Tris (pH 7.4), 150 mM NaCl, 1% Triton X-100, 1% Na deoxycholate, and 5 mM EDTA) supplemented with a protease inhibitor cocktail (Sigma) was used to obtain cell lysates by resuspending the cells in this mixture. A BCA kit (Pierce, Thermo Fisher Scientific, Waltham, MA, USA) was used to determine protein concentration in the obtained lysates, and then SDS-PAGE was performed. In brief, equal protein concentrations were heated (100 °C) for 3 min with sodium dodecyl sulfate (SDS) sample buffer (25 mL glycerol, 31.2 mL Tris buffer, 7.5 mL SDS, a dash of bromophenol blue/100 mL) and run on 10% SDS polyacrylamide gel electrophoresis (SDS-PAGE). Migrated proteins were then blotted onto PVDF membranes (Immobolin P, Watford, UK). Blocking was performed by incubating the membranes in a blocking solution (5% nonfat milk in PBS) for 1 h. Membranes were incubated with primary antibody (anti-human CLU mAb at dilution of 1:1000) overnight. The membranes were then washed three times for 10 min in TBS (0.1% Tween-20 in PBS) followed by incubation for 1 h at room temperature with horseradish peroxidase-linked IgG (1:2000 dilution in T-TBS), followed by another three washes (10 min each) with TTBS. ECL reagent (Amersham, CA, USA) was used to develop the signals on the membranes which were then imaged by Chemidoc MP imaging system (BioRad; Hercules, CA, USA)

### 2.12. DAPI Staining

Two days after the co-transfection of SNU449 cells with the circRNA-expressing vector and GFP-expressing vector, cells were fixed with 4% paraformaldehyde and permeabilized with Triton X (1%) for 4 min. The cells were then washed three times with PBS and stained with DAPI (300 nM) for 10 min. Imaging was performed by an Olympus fluorescence microscope.

### 2.13. Statistical Analysis

All experiments were performed in triplicate, and numerical data were subjected to an independent-sample *t*-test or one-way ANOVA Kruskal–Wallis test. Mann–Whitney was used for comparison of RNA data (2^−∆∆Ct^) with nonparametric data. The levels of significance were set at **** *p* ≤ 0.0001, *** *p* ≤ 0.001, ** *p* ≤ 0.01, * 0.01 < *p* ≤ 0.05, and ns, *p* > 0.05. Statistical analyses were performed using Statistical Package for the Social Science (SPSS, Chicago, IL, USA) version 22.0 and Graph Pad Prism 8 software.

## 3. Results

### 3.1. Bioinformatics-Based Retrieval of Noncoding RNAs

To investigate the role of the circRNA–miRNA–mRNA regulatory axis in HCC, we performed an in silico networking analysis. We first searched for circRNA-associated competing endogenous RNA (ceRNA) relevant to HCC according to previous microarray studies. Firstly, we analyzed circRNA genes specific to HCC through circBase (available at http://www.circbase.org/, http://circnet.mbc.nctu.edu.tw/), Aceessed on 28 November 2019) and Circ2Traits databases (available at http://gyanxet-beta.com/circdb/, http://circnet.mbc.nctu.edu.tw/), Aceessed on 28 November 2019) and found hsa-circ-000520 (cirCtrait) = *hsa-circ-0000221* (cirBase) = VIM to be highly relevant to HCC with one of the highest ranking scores (Appendix A, Appendix A). Secondly, we analyzed circRNA–miRNA interaction databases, e.g., CircNet database (available at http://circnet.mbc.nctu.edu.tw/), Aceessed on 28 November 2019) to retrieve competing endogenous RNA specific for HCC and found hsa-mir-661 (Appendix A). Computational prediction of *hsa-circ-0000221* target miRNAs revealed 40 possible interacting miRNAs with binding sites on the *hsa-circ-0000221* transcript. The interaction of hsa-mir-661 was verified by the Circ2Trait database (Figure 1). The hsa-mir-661 interaction was further validated by performing pathway enrichment analysis using miRPath tool through mircoT-CDS [24] (set at a *p*-value threshold of 0.05). This revealed significant enrichment in the cancer metabolic pathway, Hippo signaling pathway, cGMP–PKG signaling pathway, mTOR signaling pathway, AMPL signaling pathway, TGF-beta signaling pathway, insulin signaling pathway pathways in cancer, inflammatory mediator regulation of TRP channels, focal adhesion, and Wnt signaling pathway. These signaling pathways are known to play an important role in HCC carcinogenesis.

miR-661 was predicted to be direct target of *hsa-circ-000022.* Ontology analysis of hsa-mir-661 set revealed significant enrichment in G2/M transition of the mitotic cell cycle, apoptotic signaling pathway, cellular component disassembly involved in the execution phase of apoptosis, viral process, and cell death pathway These data are in agreement with gene ontology of *hsa-circ-0000221* and its possible role in HCC. Lastly, we selected PTPN mRNA as a direct target of mir-661 on the basis of TargetScan, miRwalk databases (Appendix A) and previous literature [25] that highlighted PTPN mRNA as a tumor suppressor gene involved in focal adhesion signaling, IKK/NF-κB signaling, cell proliferation, and gastrointestinal carcinogenesis. Prediction using the TargetScan tool revealed 6959 interacting mRNAs, while prediction using miRWalk revealed 57,601 interacting mRNAs. Both predictions were filtered according to prediction scores provided by both tools, and the top 100 interacting mRNAs were further considered for gene ontology and pathway enrichment analysis. The dataset of hsa-mir-661 target mRNAs was subjected to ontology prediction through WebGestalt tool [26], which uses GOSlim for ontology, showing significant enrichment in protein- and ion-binding functions (Figure 2A).

The main ontology terms determined by GOSlim were further inspected through the Enrichr tool. The pathway enrichment through the Panther database [27] showed remarkable enrichment in the Wnt pathway, which is crucial for HCC carcinogenesis. Other pathways were also significantly enriched including cadherin pathway and p53 signaling feedback loops (Figure 2B). The inspected biological process category showed positive regulation of viral transcription, apoptotic signaling, ERBB2 signaling, chemotaxis of immune cells, and ERK1 and ERK2 cascade, which has tumor-promoting effects in HCC. Inspection of cellular components revealed enrichment in CENP-A containing chromatin mitotic spindle polar microtubule and chromatin components. Additionally, molecular function ontology showed ATPase, phosphatase, DNA and RNA polymerase, and MAP kinase activity enrichment, which is strongly involved in HCC. Among these significantly interacting mRNAs, PTPN11 was predicted as the potential component of the ceRNA containing *hsa-circ-0000221* and hsa-mir-661 in HCC.

Lastly, the STRING database was used to analyze the protein–protein interactions of PTPN11 (Figure 2C). It was interesting that PTPN11 was shown to exhibit a significant interaction with MAPK3 and EGFR, along with other less significant interactions with TP53, MTOR, JUN, and STAT1. These interactions may explain the role of PTPN11 in HCC, along with MAPK3 and EGFR, which are important players in HCC carcinogenesis as reported in the HCC pathways map in the KEGG database. These interactions were further investigated through ontology prediction on the STRING database analysis tab which showed significant enrichment in HCC-relevant biological processes including positive regulation of MAPK cascade and JAK/STAT cascade. Molecular function ontologies showed enrichment in kinase binding and protein kinase activity, while cellular component ontology showed enrichment in phosphatidylinositol 3-kinase complex and Shc–EGFR complex. KEGG pathway enrichment showed enrichment in pathways in cancer, as well as the proteoglycans in cancer pathway, Ras signaling pathway, PI3K/Akt signaling pathway, FoxO signaling pathway, TNF signaling pathway, nonalcoholic fatty liver disease (NAFLD), and hepatitis C, which is strongly associated with the development of HCC.

### 3.2. Expression of hsa-circ-0000221–miR-661–PTPN11 mRNA in Serum Samples

To practically validate our molecular network, we decided to detect the expression of the *hsa-circ-0000221*, miR-661, and PTPN11 transcripts in the sera from a panel of HCC patients, chronic hepatitis C virus (HCV)-infected patients, and control volunteers. We included chronic HCV-positive patients since HCV infection is endemic in Egypt with the highest prevalence rate in the world [28], chronic HCV infection increases the risk for HCC by 15- to 20-fold, and it is reported that half of all HCC patients have serological evidence of HCV infection [29]. These patients suffered from chronic HCV infection; however, they did not develop HCC. On the other hand, some of the patients in the HCC group were HCV-positive but with malignant transformation in the liver. Hence, it was interesting to investigate how the expression of the *hsa-circ-0000221*, miR-661, and PTPN11 transcripts would be altered in these three groups of patients. The detailed clinical data of the three studied groups are shown in Table 1. Quantitative RT-PCR data of those tested RNAs revealed that the expression of *hsa-circ-0000221* and PTNPN 11 mRNA in HCC patients was significantly (10-fold and threefold, respectively) lower compared with their expression in the control group (*p* ≤ 0.0001 and *p* ≤ 0.01, respectively) (Figure 3A,C). Moreover, the relative expression level of miR-661 was significantly elevated (22-fold) in the HCC group compared with its expression in the control group (*p* < 0.05) (Figure 3B).

To further investigate the link of these molecular network members in vitro, we overexpressed the *hsa-circ-0000221* in two representative HCC cell lines, HepG2 and SNU449. qRT-PCR of the three RNAs in HepG2 showed that the expression of *hsa-circ-0000221* increased (threefold) 2 days after transfection compared with that of day 1. Our results showed that the upregulation of *hsa-circ-0000221* was significantly associated with inhibition of miR-661 expression (about fourfold lower, *p* < 0.01) and upregulation of PTPN11 mRNA (*p* < 0.05) (Figure 4A). A similar effect was observed in SNU449 cells where again the upregulation of *hsa-circ-0000221* caused an inhibition in miR-661 expression and upregulation of the PTPN11 mRNA levels (Figure 4B). This in turn inversely correlated with the cellular viability and the number of living HepG2 cells (Figure 4B,C respectively) and SNU449 cells (Figure 4E). These results indicate that *hsa-circ-0000221* may mediate PTPN 11 upregulation and subsequently show toxic effects in HCC cells.

### 3.3. hsa-circ-0000221 Induces Apoptosis in HCC

To investigate whether the expression of *hsa-circ-0000221* is inducing necrotic or apoptotic cell death, SNU449 cells were transfected (transfection efficiency was more than 50%) with *hsa-circ-0000221* expressing plasmid or control. Cells expressing the circRNA showed cellular death, which was detected by rounding cells that detach plastic as shown by regular microscopy (Figure 5A). Furthermore, exogenous expression of this gene also significantly reduced the colony-forming ability of the SNU449 and HepG2 cells as quantified in Figure 5B,C. Representative images of the assay are shown in Appendix A. Lastly, the number of annexin V-stained cells was significantly enhanced by transient transfection with *hsa-circ-0000221* in both cell lines, indicating that the observed cell death is due to apoptosis (Figure 5D,E).

### 3.4. hsa-circ-0000221 Disturbs Cell Cycle and Induces Caspase-Dependent Apoptosis in HCC

To further investigate the mechanism through which *hsa-circ-0000221* affects the viability and induces apoptosis in HCC, we studied the cell cycle after transfection with *hsa-circ-0000221*-expressing vector or empty vector. The DNA content study indicates that SNU449 cells overexpressing *hsa-circ-0000221* are relatively accumulated in the G1 phase, and those that are released from the G2/M phase hardly re-enter the cell cycle and may face cell death as indicated by the different population in the cytograph (Figure 6A left) and the quantification performed for all repeats (Figure 6A right). The debris peak indicating dead cells appeared as a sub-G1 peak only in those cells overexpressing the circRNA but not in the mock-treated cells.

To further investigate the mechanism of cellar death by *hsa-circ-0000221*, we studied the expression of a panel of apoptosis/cell cycle-related molecules (Figure 6B). Importantly, SNU449 cells overexpressing *hsa-circ-0000221* showed the accumulation of cyclin B1, cyclin D1, and CDK4, indicating prevention of S phase entry. Moreover, typical molecular signs of apoptosis were observed including PARP1 cleavage, pro-caspase depletion, and active caspase accumulation in those cells overexpressing *hsa-circ-0000221*. Lastly, apoptosis was shown to be mitochondrial-dependent because the proapoptotic effector protein BAX was significantly upregulated in the tested cells.

## 4. Discussion

Although circRNAs were treated as transcriptional noise since they were first discovered in 1979 [30], their role in disease development and particularly in tumorigenesis of different forms of cancer has been gaining more interest [31]. For example, a database of circRNAs directly detected in prostate cancer tissues has been recently curated and termed MiOncoCirc [9]. In this study, we focused on the role of circRNAs in HCC, which may lead to a better understanding of the molecular mechanism of HCC carcinogenesis and could contribute to enhancing diagnostic and/or therapeutic strategies [32,33,34].

Our in silico investigation revealed a novel molecular network in HCC of *hsa-circ-0000221*–miR-661–PTPN11 mRNA. This molecular network, to the best of our knowledge, is reported here for the first time. We used a panel of serum samples from HCC patients, chronically infected HCV patients, and normal volunteers, of different ages and sex, to validate this network. We found that *hsa-circ-0000221* was significantly depleted in the sera of HCC patients compared to normal controls and HCV patients. These depleted levels were associated with a high expression pattern of miR-661 and low expression of PTPN11 transcripts in the serum, which is contrary to the case of patients with nonmalignant livers (both normal controls and HCV patients). These data suggest that *hsa-circ-0000221* and, consequently, the *hsa-circ-0000221*–miR-661–PTPN11 mRNA axis are deregulated specifically in malignancy and not with viral infection of the liver, even though HCV infection is a well-documented cause of HCC. Previous reports have shown that HCV can modulate the abundance of some circRNAs in infected cells [35]. Although this is not apparent for *hsa-circ-0000221* levels in serum, it would be interesting in the future to investigate whether its expression is affected in liver cells upon HCV infection. This is the first report showing a role for *hsa-circ-0000221* and its deregulation in cancer. The *hsa-circ-0000221*–miR-661–PTPN11 mRNA axis was validated to be present in both HCC cell lines tested, indicating that it could be a key player in HCC pathogenesis.

miR-661 can be oncogenic or tumor-suppressive in cancer development [36,37]. It has been reported to be deregulated in breast cancer [38] and ovarian cancer [39], while a recent report showed its role in HCC, where a significant correlation was found between the levels of miR-661 and the expression of NF-κB-p65 and hepatitis B virus transactivator protein (HBx) in a human HCC cell line through HBx-mediated stimulation of metastasis-associated protein 1 (MTA1) [40].

According to our findings, miR-661 has an oncogenic function in HCC as it targets PTPN11; depletion of miR-661 may give the chance for PTPN11 transcript to increase and subsequently inhibit cellular proliferation in HCC. Our results proved that miR-661 could be depleted indirectly via upregulation or overexpression of *hsa-circ-0000221* in HCC cells. Furthermore, our finding indicates that miR-661 depletion and the subsequent increase in PTPN11 are a consequence of *hsa-circ-0000221* overexpression in vitro, which is associated with retardation of the cellular viability and/or apoptosis induction.

PTPN11/Shp2 is an intracellular tyrosine phosphatase with two Src-homology 2 (SH2) domains [25] that acts to promote activation of the Ras/ERK pathway by cytokines, growth factors, and hormones. PTPN11 mutations have been linked to juvenile myelomonocytic leukemia (JMML) [41], colorectal cancer [42], breast cancer [43], and HCC [44]. The tumor-suppressive role of PTPN11 in HCC is mediated through the IκB kinase/NF-κB (IKK/NF-κB) pathway through promoting IL-6-stimulated Stat3 activation [45,46]. Thus, the *hsa-cir-0000221*–miR-661–PTPN11 mRNA axis could be working through the IKK/ NF-κB pathway to inhibit cancerous proliferation.

Interestingly, other circular RNAs and microRNAs act as post-transcriptional modifiers of genes linked to the IKK/NF-κB signaling pathway in the liver [47]. Moreover, dysregulation of the IKK–NF-κB–IL-6 signaling pathway has been linked to the development of HCC [48,49].

Our study uncovered the possible effect of *hsa-circ-0000221* on HCC. The therapeutic efficacy of exogenously expressed *hsa-circ-0000221* in HCC cell lines was also evaluated. Our results showed that the upregulated *hsa-circ-0000221* expression is significantly associated with inhibition of miR-661 expression and upregulation of PTPN11 mRNA, which in turn inversely correlated with HCC cell viability and induced mitochondrial-dependent and caspase-dependent apoptosis. However, it needs to be noted that those phenotypic effects in HCC cells cannot be attributed directly to changes in miR-661 or PTPN11 levels as this was not experimentally validated in our study. Further investigation involving miRNA mimics or PTPN11 knockdown should be performed in future studies. CircRNAs are well known for being more stable than other RNA species in the cellular cytoplasm, exhibiting longer half-lives due to their covalently closed circular structure. They are resistant to RNase R and other exonucleases [50]. Moreover, due to this feature, circRNAs are very prevalent in blood, serum, and other body fluids such as interstitial fluid, saliva, and cerebrospinal fluid, making them ideal molecular biomarkers for several diseases [51]. Robust identification and quantification of their levels in patient samples by next-generation sequencing technologies have been shown [51,52,53]. Moreover, targeting miRNAs in vivo in HCV-infected nonhuman primates has been established [54], and the use of miRNA antagomirs for cancer treatment has been shown [55].

On the basis of the aforementioned findings, we adopted an integrative approach combining in silico data analysis with clinical and experimental validation. A reasonable interpretation of our hypothesis is that *hsa-circ-0000221* competes with miR-661 and probably sponges miR-661 to modulate the expression of PTPN11 mRNA that is closely linked to the IκB kinase/NF-κB signaling pathway, which in turn affects the apoptosis pathway in the liver [56].

Taken together, we think that the depletion of *hsa-circ-0000221* in the HCC patients’ sera is a stable serum biomarker for HCC diagnosis with high resistance to the exonuclease RNase and a potential candidate for translation to be an effective therapeutic target, thereby improving the prognosis and survival rate of HCC patients. Introducing *hsa-circ-0000221* into HCC cells or tissues may be a promising therapeutic strategy (Figure 7).

## 5. Conclusions

The current study could lay the foundation for revealing the crucial roles of *hsa-circ-0000221* in HCC development and prognosis, paving the way toward better diagnostic and/or therapeutic strategies for better HCC management. This study was limited by its small sample size as an initial screening group, which was not sufficiently powered to examine the diagnostic efficacy of the chosen biomarkers. Therefore, further large multicenter studies are strongly recommended to evaluate the performance characteristics of the chosen RNA network. More in vitro mechanistic studies are need to highlight the underlying in-depth mechanism beyond the role of the chosen circular RNA network in HCC due to complexity of the circRNA intracellular mechanism of action.

## Figures and Tables

**Figure 1 pharmaceutics-14-00138-f001:**
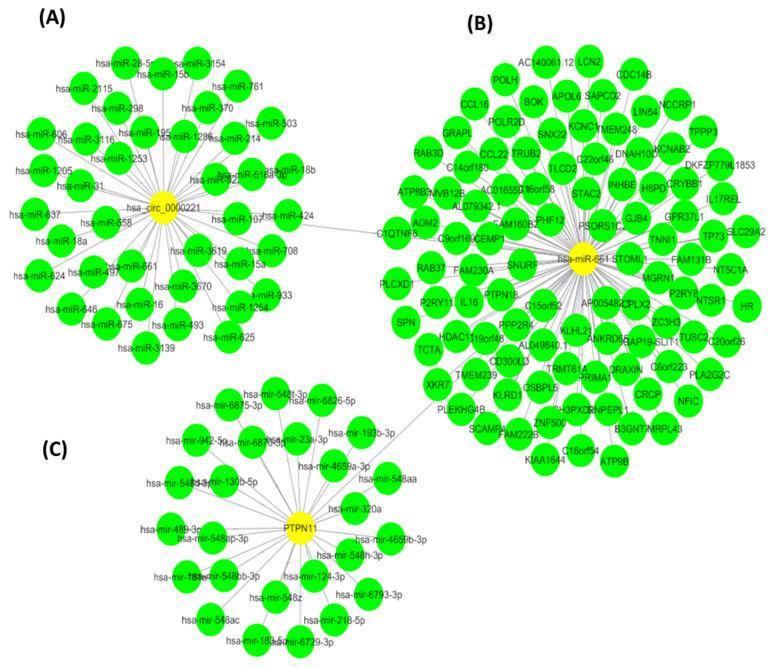
Computationally determined *hsa-circ-0000221* ceRNA network. The network was constructed using Cytoscape software (Version 3.5.1) and shows the (**A**) predicted miRNAs targeted by *hsa-circ-0000221*, (**B**) the mRNAs targeted by miRNA hsa-mir-661, and (**C**) the network of miRNAs regulating PTPN11. The lengths of edges in this network correspond to the significance of the interaction between network nodes.

**Figure 2 pharmaceutics-14-00138-f002:**
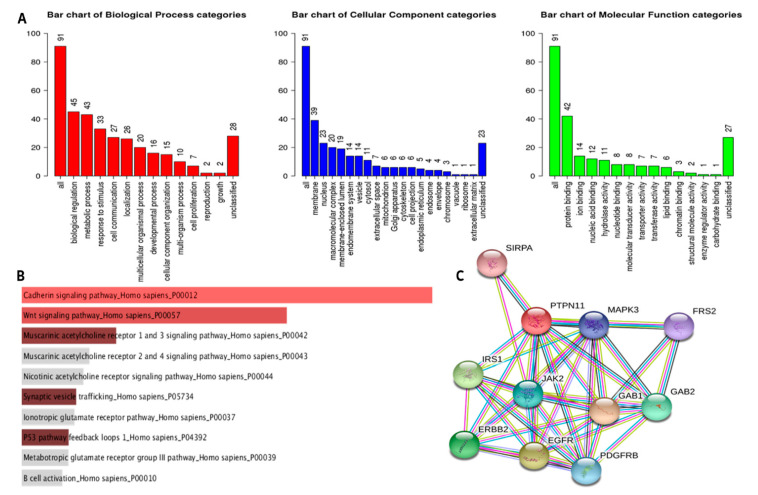
Gene ontology prediction for the *hsa-circ-0000221* ceRNA network. (**A**) GOSlim ontology prediction summary for top 100 mRNA targets of hsa-mir-661 as predicted by the WebGestalt server. The three bar charts represent the main categories of gene ontology including biological process (red), cellular component (blue), and molecular function (green). (**B**) Significant Panther pathways predicted in which mRNA targets of hsa-mir-661 were enriched, as predicted using Enrichr. (**C**) STRING database network of PTPN11 significant protein–protein interactions.

**Figure 3 pharmaceutics-14-00138-f003:**
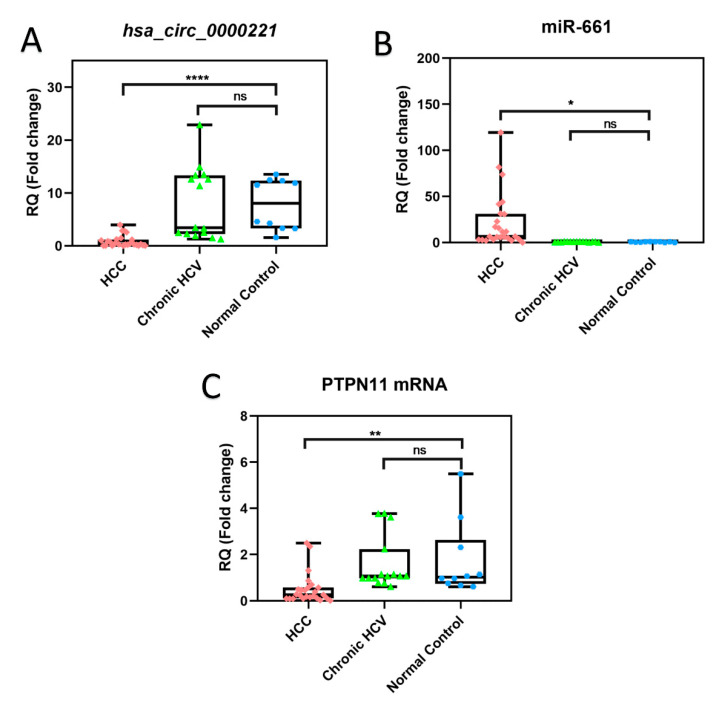
Expression of serum *hsa-circ-0000221*, miR-661, and PTPN 11 mRNA as determined by qRT-PCR in HCC, chronic HCV, and healthy individuals. The boxplots show the expression of serum (**A**) *hsa-circ-0000221*, (**B**) miR-661, and (**C**) PTPN11 mRNA among three patient groups: HCC, chronic HCV, and normal control. Individual data points represent individual patients in each group, while error bars indicate SD. Statistics were performed using Student’s unpaired *t*-test; **** *p* ≤ 0.0001, ** *p* ≤ 0.01, * 0.01< *p* ≤ 0.05, and ns, *p* > 0.05.

**Figure 4 pharmaceutics-14-00138-f004:**
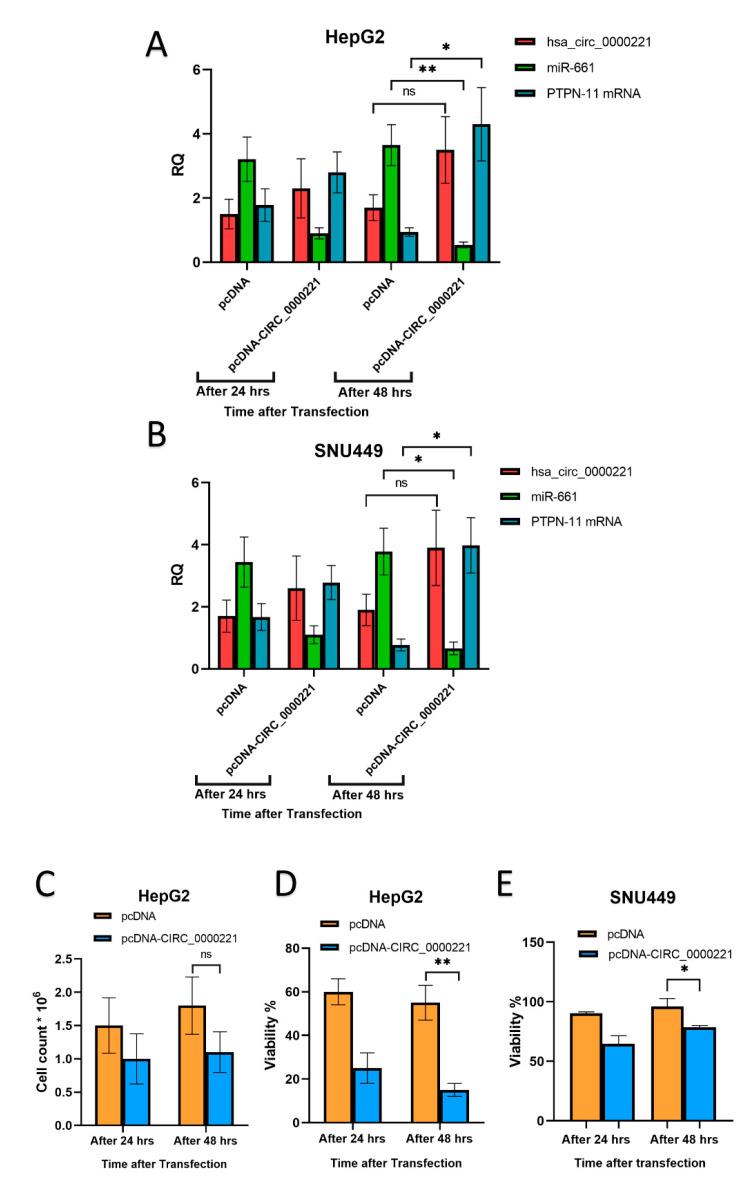
Effect of *hsa_circ_0000221* overexpression in HepG2 and SNU449 cells. *hsa_circ_0000221*, miR-661, and PTPN 11 mRNA were quantified by qRT-PCR 1 and 2 days after transfection of either pcDNA-CIRC-0000221 or control vector in (**A**) HepG2 and (**B**) SNU449 cell lines. Data for each gene were normalized to their counterparts in cells at 0 h. (**C**) Cell count in HepG2 cells, and cellular viability in (**D**) HepG2 and (**E**) SNU449 cells were measured 1 and 2 days after transfection. Error bars represent SD. Statistics were performed using Student’s unpaired *t*-test; ** *p* ≤ 0.01, * 0.01 < *p* ≤ 0.05, and ns, *p* > 0.05.

**Figure 5 pharmaceutics-14-00138-f005:**
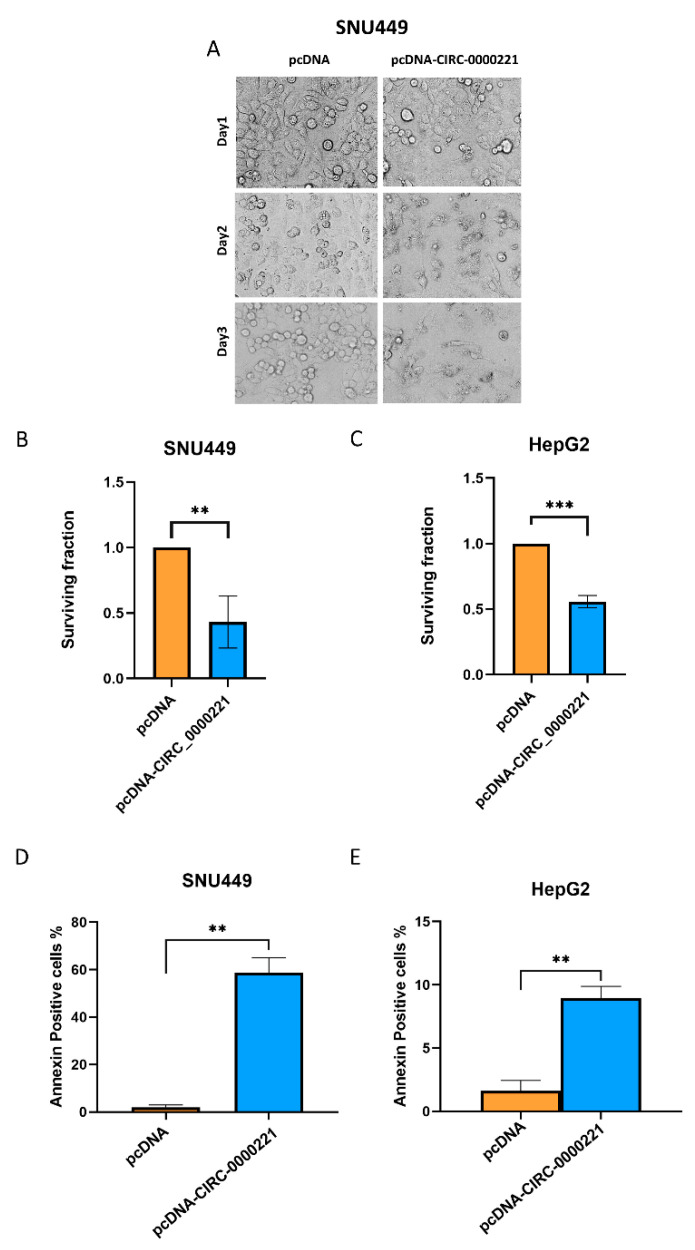
Exogenous expression of *hsa-circ-0000221* induces apoptosis in HepG2 and SNU449 cells. (**A**) Phase contrast of SNU449 cells 1, 2, and 3 days after transient transfection with *hsa-circ-0000221*-expressing vector or control. Clonogenic survival assay of (**B**) SNU449 and (**C**) HepG2 cells after overexpression of *hsa-circ-0000221* or empty vector reflecting the differential colony-forming ability between the two conditions as indicated by the surviving fraction. Annexin-positive cell percentage as determined by flow cytometry of (**D**) SNU449 and (**E**) HepG2 cells after overexpression of *hsa-circ-0000221* or empty vector following annexin V and propidium iodide staining. Statistics were performed using Student’s unpaired *t*-test; ** *p* ≤ 0.01, *** *p* ≤ 0.001.

**Figure 6 pharmaceutics-14-00138-f006:**
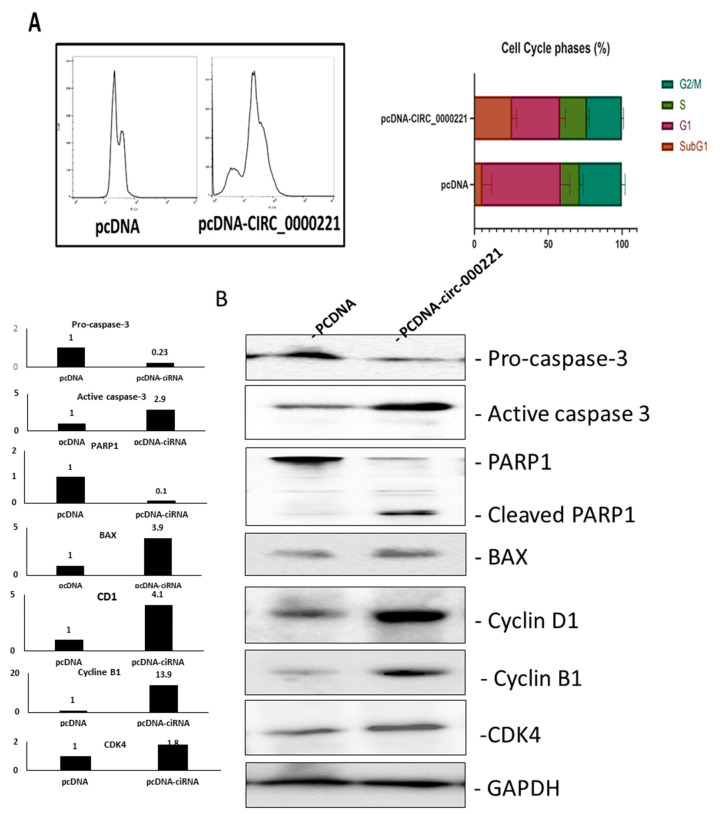
*hsa-circ-0000221* overexpression induces cell-cycle arrest and caspase-dependent apoptosis. (**A**) Representative FACS analysis of SNU449 cells after transient transfection with *hsa-circ-0000221-*expressing vector or backbone (**left**). Quantification of cell-cycle phases in each condition (**right**). The cells overexpressing *hsa-circ-0000221* showed accumulation in the G1 phase compared with mock. Error bars represent SD. (**B**) Western blotting analysis (**left**) of the same two clones as above, showing the effect of *hsa-circ-0000221* overexpression on procaspase-3, caspase-3, PARP1, CB1, CCDN1, CDK4, and Bax. GAPDH was also detected as a loading control. Quantification of the bands for each blot and each condition is shown (**right**). AU: arbitrary units.

**Figure 7 pharmaceutics-14-00138-f007:**
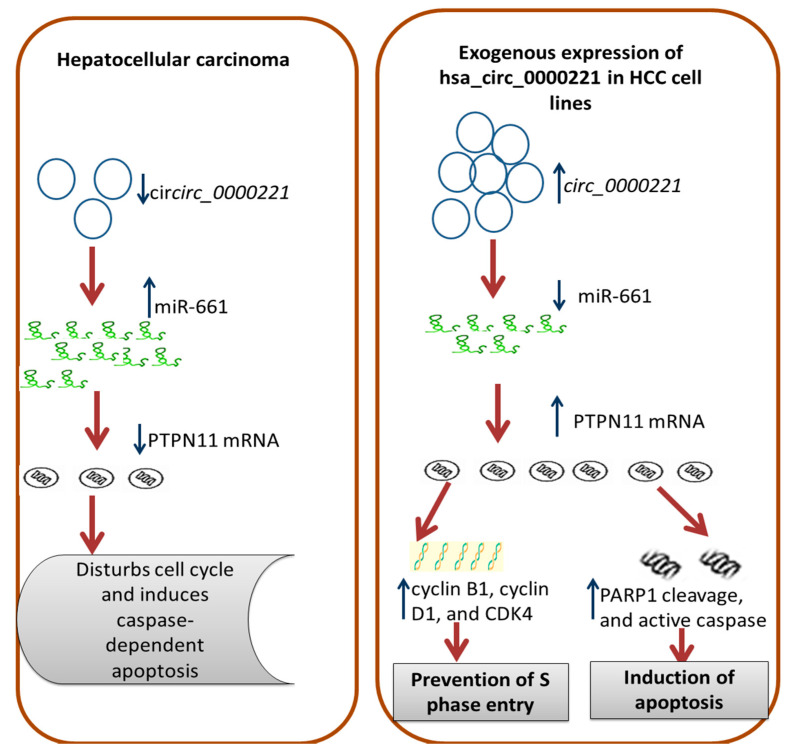
Schematic figure summarizing the role of hsa-circ-0000221–miR-661–PTPN11 mRNA in HCC.

**Table 1 pharmaceutics-14-00138-t001:** Study population demographic and clinical characteristics (*N* = 50).

	Malignant (*n* = 25)	CHC (*n* = 15)	Normal (*n* = 10)	*p*	χ^2 (a)^ F ^(b)^
Age					
≥57 years (*n* = 29)	14 (56%)	10 (66.7%)	5 (50%)	0.68	χ^2 (a)^ = 0.76
<57 years (*n* = 21)	11 (44%)	5 (33.3%)	5 (50%)		
Sex:					
Male (*n* = 27)	16 (66%)	4 (26.7%)	7 (70%)	0.051	χ^2 (a)^ = 6.5
Female (*n* = 23)	9 (34%)	11 (73.3%)	3 (30%)		
Smoking:					
Smoker (*n* = 17)	9 (36%)	6 (40%)	2 (20%)	0.56	χ^2 (a)^ = 0.125
Nonsmoker (*n* = 33)	16 (64%)	9 (60%)	8 (80%)		
HCV antibodies:					
Positive (*n* = 35)	21 (84%)	14 (93.3%)	0 (0%)	0.000 **	χ^2 (a)^ = 17.5
Negative (*n* = 15)	4 (16%)	1 (6.7%)	10 (100%)		
HBV sAg:					
Positive (*n* **=** 1)	1 (4%)	0 (0%)	0 (0%)	0.061	χ^2 (a)^ = 1.596
Negative (*n* = 49)	24 (96%)	15 (100%)	10 (100%)		
Cirrhosis:					
Cirrhotic (*n* = 29)	21 (84%)	8 (53.3%)	0 (0%)	0.000 **	χ^2 (a)^ = 13.1
Noncirrhotic (*n* = 21)	4 (16%)	7 (46.7%)	10 (100%)		
AST	75.6 ± 43.66	68.4 ± 18.4546	38.06 ± 18.1	0.000 **	F ^(b)^ = 19.7
ALT	54.1 ± 46.2	39.1 ± 17.27	29.3 ± 11.43	0.000 **	F ^(b)^ = 17.2
Albumin	2.74 ± 0.31	3.31 ± 0.22	3.6 ± 0.5	0.000 **	F ^(b)^ = 45.3
Total Bilirubin	4.9 ± 0.57	3.8 ± 2.76	1.38 ± 1.7627	0.000 **	F ^(b)^ = 24.5
Direct Bilirubin	2.89 ± 0.55	2.6 ± 3.4	0.73 ± 1.2	0.000 **	F ^(b)^ = 11.05
INR	1.4 ± 0.3	1.3 ± 0.095	1.14 ± 0.15	0.000 **	F ^(b)^ = 57.3
α-feto-protein	3193.90 ± 485.7	525.52 ± 122.3	165.66 ± 2.2	0.000 **	F ^(b)^ = 11.7
Child score		-	-	-	-
A5	4 (16%)				
A6	3 (12%)				
B6	6 (24%)				
B7	1 (4%)				
B8	5 (20%)				
C10	6 (24%)				
Mean size of the tumor		-	-	-	-
≥3 cm	18 (72%)				
<3 cm	7 (28%)				
BCLC stage		-	-	-	-
A	23 (92%)				
D	2 (8%)				

^a^ Chi-square test; ^b^ one-way ANOVA test; ** highly significant correlation was detected between investigated groups at *p* < 0.01. HCV: hepatitis C virus, CHC: chronic HCV infection, HBV sAg: hepatitis B virus surface antigen, AST: aspartate transaminase, ALT: alanine transaminase, INR: international normalized ratio, BCLC: Barcelona clinic liver cancer.

## Data Availability

The data reported in this study are available on request from the corresponding authors.

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
