# Peer review of "Impact of circ-0000221 in the Pathogenesis of Hepatocellular via Modulation of miR-661–PTPN11 mRNA Axis"

_pharmaceutics, 2022, doi:10.3390/pharmaceutics14010138_

Round 1
Reviewer 1 Report
The title is interesting and it would attract readers from various fiends including cancer therapy and those working in field of gene therapy. Understanding interactions among circRNAs and miRNAs are of importance determining progression of hepatocellular carcinoma cells. The miRNAs and circRNAs discussed in this current manuscript are novel and have not been examined before. I suggest publication of this article after minor revision as following:
- The title need to be revised and make it more attractive for readers.
- A schematic figure need to be added discussing molecular pathways involved in hepatocellular progression.
- Conclusion should be improved and elaborated by adding more information about limitations of current works and providing directions for future studies.
- There are a few references from 2020 and 2021 and references should be updated to improve quality and visibility of their work.
- The second paragraph of introduction is discussing circRNAs. However, information are not enough and complete and should be improved. Suggested article (Doi, 10.1016/j.canlet.2021.03.025).
- The first paragraph of introduction is about hepatocellular carcinoma. Authors need to provide more information about drug resistance feature of cancer cells, their progression (proliferation and invasion) and pathways involved in their malignancy.
Author Response
Reviewer 1
The title is interesting and it would attract readers from various fiends including cancer therapy and those working in field of gene therapy. Understanding interactions among circRNAs and miRNAs are of importance determining progression of hepatocellular carcinoma cells. The miRNAs and circRNAs discussed in this current manuscript are novel and have not been examined before. I suggest publication of this article after minor revision as following:
- Thanks very much for your positive encouraging comments
- The title need to be revised and make it more attractive for readers. pathogenesis
- As suggested by the reviewer, the title has been modified as follows:
- Impact of circ_0000221_ in the Pathogenesis of Hepatocellular Via ,Modulation of miR-661__PTPN11 mRNA Axis
- A schematic figure need to be added discussing molecular pathways involved in hepatocellular progression.
- As suggested by the reviewer, a schematic figure summarizing the role of hsa_circ_0000221_miR-661_PTPN11 mRNA in HCC
- Conclusion should be improved and elaborated by adding more information about limitations of current works and providing directions for future studies.
- As suggested by the reviewer: study limitations have been added to the conclusion as follows:
- Finally, this study was limited by small sample size as an initial screening group which was not sufficiently powered to examine the diagnostic efficacy of the chosen biomarkers. Therefore, further large multicenter studies are strongly recommended to evaluate the performance characteristic of the chosen RNA network. More in vitro mechanistic studies are need to highlight underlying molecular mechanism beyond the role of the chosen circular RNA network in HCC due to complexity of CircRNA intracellular mechanism of action
- There are a few references from 2020 and 2021 and references should be updated to improve quality and visibility of their work.
- As suggested by the reviewer, some references has been updated as follows:
- Ali HS, Boshra MS, El Meteini MS, Shafei AE, Matboli M. lncRNA- RP11-156p1.3, novel diagnostic and therapeutic targeting via CRISPR/Cas9 editing in hepatocellular carcinoma.Genomics. 2020 Sep;112(5):3306-3314.
- Rashed, W.M., Kandeil, M.A.M., Mahmoud, M.O. et al. Hepatocellular Carcinoma (HCC) in Egypt: A comprehensive overview. J Egypt Natl Canc Inst 32, 5 (2020). https://doi.org/10.1186/s43046-020-0016-x.
- Zhou XY, Yang H, Bai YQ, Li XL, Han SY, Zhou BX. hsa_circ_0006916 promotes hepatocellular carcinoma progression by activating the miR-337-3p/STAT3 axis. Cell Mol Biol Lett. 2020;25(1):47. Published 2020 Nov 2. doi:10.1186/s11658-020-00238-5.
- Hussen BM, Honarmand Tamizkar K, Hidayat HJ, Taheri M, Ghafouri-Fard S. The role of circular RNAs in the development of hepatocellular carcinoma. Pathol Res Pract. 2021 Jul;223:153495. doi: 10.1016/j.prp.2021.153495. Epub 2021 May 24. PMID: 34051512.
- Jin J, Liu H, Jin M, Li W, Xu H, Wei F. Silencing of hsa_circ_0101145 reverses the epithelial-mesenchymal transition in hepatocellular carcinoma via regulation of the miR-548c-3p/LAMC2 axis. Aging (Albany NY). 2020 Jun 18;12(12):11623-11635. doi: 10.18632/aging.103324. Epub 2020 Jun 18. PMID: 32554866; PMCID: PMC7343517.
- Cheng J, Luan J, Chen P, Kuang X, Jiang P, Zhang R, Chen S, Cheng F, Gou X. Immunosuppressive receptor LILRB1 acts as a potential regulator in hepatocellular carcinoma by integrating with SHP1. Cancer Biomark. 2020;28(3):309-319. doi: 10.3233/CBM-190940. PMID: 32390601.
- Ely A, Bloom K, Maepa MB, Arbuthnot P. Recent Update on the Role of Circular RNAs in Hepatocellular Carcinoma. J Hepatocell Carcinoma. 2021;8:1-17. Published 2021 Jan 27. doi:10.2147/JHC.S268291
- The second paragraph of introduction is discussing circRNAs. However, information are not enough and complete and should be improved. Suggested article (Doi, 10.1016/j.canlet.2021.03.025).
- As suggested by the reviewer, more details about cirRNAs role in cancer using the suggested article has been added as follows:
- circRNAs may act as competing endogenous RNA (ceRNA) via modulating miRNA expression leading to NF-κB activation that may results in cancer progression and also mediates chemoresistance(Mirzaei et al.,2021).
- The first paragraph of introduction is about hepatocellular carcinoma. Authors need to provide more information about drug resistance feature of cancer cells, their progression (proliferation and invasion) and pathways involved in their malignancy
- As suggested by the reviewer, more information about drug resistance feature of cancer cells, their progression (proliferation and invasion) and pathways involved in their malignancy has been added to introduction as per your valuable advice.
- circRNA-SORE (a circular RNA upregulated in sorafenib-resistant HCC cells), play crucial role in maintenance and spread of sorafenib resistance(Xu et al.,2020).
- Many HCC patients respond poorly to sorafenib (the first-line chemotherapeutic therapy for advanced HCC) and develop drug resistance after several months of treatment. Chemotherapeutic resistance in HCC involves different mechanisms e.g. , epithelial−mesenchymal transition, cancer stem cells, autophagy, and epigenetic regulation(Wei et al,m2019). Several signaling pathways ar were enrolled in chemotherapeutic resistance in HCC as; TNFα/NF-κB, Wnt/β-catenin, TGFβ, Ras/MEK/ERK. And JAK/STAT pathways (Xu et al.,2020).

Reviewer 2 Report
The aim is stated clear. The authors stated clearly what study found and how they did it.
The title is informative and relevant.
The references are relevant and recent. The cited sources are referenced correctly. Appropriate and key studies are included.
The introduction reveals what is already known about this topic. The research question is clearly outlined. The research question also justified given what is already known about the topic.
The process of selection of the subjects was clear. The variables are well defined and measured appropriately. The study methods are valid and reliable. There are enough details provided in order to replicate the study.
The data is presented in an appropriate way. The text in the results add to the data and it is not repetitive. Statistically significant results are clear. It is clear which results are with practical meaning. Results are discussed from different angles and placed into context without being overinterpreted.
The conclusions answer the aim of the study. The conclusions are supported by references and own results.
Specific comments on weaknesses of the article and what could be improved:
Major points
- Please, state the limitations of the study
- Add a passage in the discussion on how these results would be crucial for the practice
Minor points - some typos should be corrected
Author Response
Reviewer 2
The aim is stated clear. The authors stated clearly what study found and how they did it.
The title is informative and relevant.
The references are relevant and recent. The cited sources are referenced correctly. Appropriate and key studies are included.
The introduction reveals what is already known about this topic. The research question is clearly outlined. The research question also justified given what is already known about the topic.
The process of selection of the subjects was clear. The variables are well defined and measured appropriately. The study methods are valid and reliable. There are enough details provided in order to replicate the study.
The data is presented in an appropriate way. The text in the results add to the data and it is not repetitive. Statistically significant results are clear. It is clear which results are with practical meaning. Results are discussed from different angles and placed into context without being overinterpreted.
The conclusions answer the aim of the study. The conclusions are supported by references and own results.
Specific comments on weaknesses of the article and what could be improved:
- Thanks very much for your positive encouraging comments
Major points
- Please, state the limitations of the study
- As suggested by the reviewer: study limitations have been added to the conclusion as follows:
- Finally, this study was limited by small sample size as an initial screening group which was not sufficiently powered to examine the diagnostic efficacy of the chosen biomarkers. Therefore, further large multicenter studies are strongly recommended to evaluate the performance characteristic of the chosen RNA network. More in vitro mechanistic studies are need to highlight underlying molecular mechanism beyond the role of the chosen circular RNA network in HCC due to complexity of CircRNA intracellular mechanism of action.
- Add a passage in the discussion on how these results would be crucial for the practice
- As suggested by the reviewer: a passage in the discussion on how these results would be crucial for the practice has been added as follows;
- Taken together, we think that the depletion of hsa_circ_0000221 in the HCC patients’ sera is a stable serum biomarker for HCC diagnosis with high resistance to the exonuclease RNase and potential candidates for translation to be an effective therapeutic target, to improve the prognosis and survival rate of HCC patients.
- Minor points - some typos should be corrected
- As suggested by the reviewer: typos all through MS has been edited.
